# Optimized the Microgrid Scheduling with Ice-Storage Air-Conditioning for New Energy Consumption

**Yiping Xiao** [1,2]**, Jiaxuan Li** [1,2,*]**, Xiyao Gong** [1,2] **and Jun He** [1,2]

1    School of Electrical and Electronic Engineering, Hubei University of Technology, Wuhan 430068, China; yp_x@hbut.edu.cn (Y.X.); 102210301@hbut.edu.cn (X.G.); hejun@hbut.edu.cn (J.H.)
2    Hubei Key Laboratory for High-Efficiency Utilization of Solar Energy and Operation Control of Energy Storage System, Hubei University of Technology, Wuhan 430068, China
*    Correspondence: 102200182@hbut.edu.cn

**Abstract:** In the face of the stochastic, fluctuating, and intermittent nature of the new energy output, which brings significant challenges to the safe and stable operation of the power system, it is proposed to use the ice-storage air-conditioning to participate in the microgrid optimal scheduling to improve wind and light dissipation. This paper constructs an optimal scheduling model for the ice-storage air-conditioning to participate in the microgrid, analyzes the regulation advantages of the ice-storage air-conditioning's cold storage and cold release process and its participation in the scheduling process; secondly, based on the scenario method, the scenario modeling of the microgrid distributed wind and light output uncertainty and cold load uncertainty is carried out; lastly, an optimal scheduling model is constructed to minimize the operation and maintenance cost of each unit and the storage cost of the ice-storage air-conditioning, the highest rate of clean energy generation and the lowest cost of electricity consumption of the air-conditioning users as the objective function. The example simulations show that the proposed optimal scheduling model can promote the new energy consumption rate of the microgrid, proving that the ice-storage air-conditioning is more economical compared with ordinary air-conditioning and that the operating cost within the optimized microgrid is reduced by about 10.5%. The cost of the air-conditioning users' electricity consumption has been reduced by about 11.7% after responding to the regulation.

**Keywords:** new energy consumption; ice-storage air-conditioning; microgrid scheduling; the NSGA-III algorithm

## 1. Introduction

The large-scale development and utilization of renewable energy has become the most significant trend of today's energy development. As a new distributed energy organization, the microgrid fully complies with the characteristics of resource distribution and power demand, effectively distributing power generation efficiency. However, the randomness, volatility, and intermittency of new energy output bring significant challenges to the safe and stable operation of the power system. The mismatch between the output characteristics of wind, photovoltaic, and load characteristics will lead to the power supply of the period [1]; therefore, mobilizing demand-side resources to participate in the regulation of the microgrid is very important to improve the rate of new energy consumption.

Along with the development of the economy and industry, the total social power load demand is increasing, and the summer peak load is rising year by year, in which the air-conditioning load is synchronized with the peaks and valleys of social power consumption, which further increases the peak-valley difference of the power grid [2]. The ice-storage air-conditioning system uses the nighttime trough power load to make ice storage cold. It melts ice into water to release the cold stored in ice during the daytime peak hours [3] so that the power grid can be shifted to peak operation and play the role of "cutting peaks and

filling valleys". Therefore, studying the optimal scheduling of ice-storage air-conditioning systems has significant social and economic benefits [4,5].

Numerous academics have examined the Ice-Storage Air-Conditioning as an excellent demand-side adjustable resource. Ref. [6] proposes a novel photovoltaic direct-drive ice-storage air-conditioning system that fully utilizes the photovoltaic power in the microgrid for ice storage. Ref. [7], a data-based Adaptive Approximate Dynamic Programming (ADP) method is used to solve the optimal control scheme of the ice-storage air-conditioning system, which can satisfy the needs of melting the ice when the load demand is high and utilizing the cold storage function to deploy its power consumption mode flexibly. Ref. [8] for the growing cold load of the plant, the use of the Mixed Integer Programming (MIP) model to optimize the scheduling of chiller units through the fan and pump operation mode to simplify the process of chiller operation has improved the economy at the same time as the reliability and usability of the model have been improved to achieve the optimization of the scheduling of the model to provide the basis for the model. Subsequently, to address the accuracy problem of small-scale scheduling of chiller units, ref. [9] establishes a chiller model based on the feature recognition method to assign the optimal cooling loads for chiller units with different efficiencies and achieve high accuracy, providing new ideas for modeling chiller units. Ref. [10] focuses on operating Chilled Water Storage (CWS) systems for optimal scheduling. Ref. [2] proposes a water and wind disposal scheduling method based on an ice-storage air-conditioning system, which controls the ice storage capacity, the number of controllable air conditioning units, and the ice storage margin, realizes the optimal scheduling of generating units, mitigates the problem of power imbalance, reduces the peak-valley difference of power loads, improves the system's operation efficiency, and saves users' electricity costs. Ref. [11] investigates the application of ice storage devices in a nuclear power plant building. It constructs a dual-time-scale HVAC optimization model containing ice storage that reduces energy consumption. Ref. [12] proposes an optimal scheduling model for the ice-storage air-conditioning considering dynamic tariff demand response, which is solved by using the improved Ripple Bee Swarm Optimization (IRBSO) algorithm to alleviate the power imbalance problem, reduce the peak-to-valley difference of the power loads, improve the system operation efficiency, and save the user's electricity cost. Ref. [13] a cold load demand-side management strategy is proposed to reduce investment and losses in battery storage systems by applying ice-storage air-conditioning to the microgrid.

This paper proposes a multi-objective optimization model based on the NSGA-III algorithm and user comfort for the ice-storage air-conditioning to participate in wind power consumption. Considering the demand response of user comfort, the optimization model with the objectives of maximum wind power and light consumption, minimum operation cost, and minimum user cost is established, and the NSGA-III algorithm is used to solve the model.

## 2. The Ice-Storage Air-Conditioning Modeling

Refrigeration equipment, ice storage equipment, and users are the main components of ice-storage air-conditioning, which can be divided into an ice production and storage phase and an ice melting and refrigeration phase according to its mode of operation [14]. Ice storage refrigeration is the ice-storage air-conditioning that produces the cold volume in the form of ice storage and generally occurs in the night-time users of electricity in the low valley period; ice refrigeration is the ice-storage air-conditioning that, through the melting of ice and refrigeration operation, the release of cold volume for the users of cold volume generally occurs during the daytime users of electricity in the flat section and the peak section. This control strategy reflects the load transfer characteristics of the ice-storage air-conditioning. For the grid side, this control strategy helps to reduce the peak and valley differences generated when the power supply of electric equipment increases, easing the pressure on the power supply of the grid; for the user side, this control strategy helps to

reduce the user's price of electricity, which improves the economy of the air-conditioning users of electricity (Figure 1).

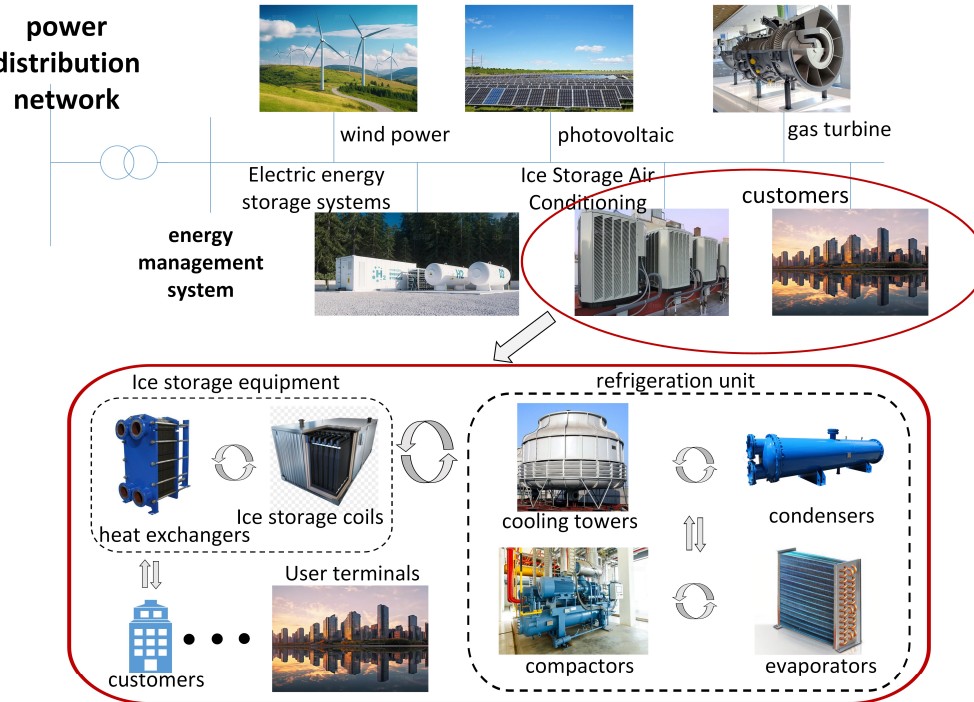

**Figure 1.** The architecture of the ice-storage air-conditioning participates in the microgrid's distributed power consumption.

### 2.1. Ice Storage Process

The ice storage tank can only melt or refrigerate during the same period. The two processes cannot be carried out simultaneously, so the ice-storage air-conditioning will melt ice and refrigerate to the power of 0 during the ice-storage process at night. The chiller will satisfy all the system's cold power demands. The relationship between the refrigeration electric power, the ice-producing electric power, and the cold power can be represented as follows: during the nighttime ice storage process, the power of the ice-storage air-conditioning is equal to the power of the chiller and the ice maker.

$$\begin{cases} P_{Bcin}(t) = P_{Bc1in}(t) + P_{Bc2in}(t) \\ P_{Bc1in}(t) = P_{Be1}(t) \cdot COP_{Be1} \\ P_{Bc2in}(t) = 0 \end{cases} \tag{1}$$

where: $P_{Bcin}(t)$ is the cold power (kw) of the ice-storage air-conditioning; $P_{Bc1in}(t)$ is the cold power (kw) of the ice storage tank under the ice-making condition; $P_{Be1}(t)$ is the electric power (kw) consumed by the ice storage tank to store ice for refrigeration; and $COP_{Be1}$ is the energy-efficiency coefficient of the ice-storage air-conditioning in the ice-making condition, which describes the refrigeration performance of the ice-storage air-conditioning in the ice-making condition and is the ratio of the refrigeration capacity and the power consumption.

The amount of cold stored in the ice-storage air-conditioning at time *t* is:

$$\begin{cases} Q_{Cooling}(t) = Q_{Cold}(t) + Q_{Ice}(t) \\ Q_{Cold}(t) = Q_{Cold}(t-1)(1 - \sigma_{Bc}) + [P_{Bc1in}(t)\eta_{Bcin} - P_{Bc1out}(t)/\eta_{Bcout}]\Delta(t) \\ Q_{Ice}(t) = 0 \end{cases} \tag{2}$$

where: $Q_{Cooling}(t)$ is the total cooling capacity (kw·h) of the ice-storage air-conditioning at time *t*; $Q_{Cold}(t)$ and $Q_{Cold}(t-1)$ are the cold capacity (kw·h) stored in the ice storage tank

at time $t$ and time $t-1$; $\Delta(t)$ is the time interval, which is 1 h in this study; $P_{Bc1in}(t)$ is the cooling power (kw) of the ice storage tank at time $t$; $P_{Bc1out}(t)$ is the cooling power (kw) of the ice storage tank at time $t$; $\sigma_{Bc}$, $\eta_{Bcin}$ and $\eta_{Bcout}$ are the cold loss rate of the ice storage tank, the cold storage efficiency and the cooling efficiency of the ice storage tank.

## 2.2. Ice-Melting Process

In the ice-melting process of the ice-storage air-conditioning, the sum of the power of the refrigerator and the ice machine constitutes the power of the ice-storage air-conditioning, and the relationship between the refrigeration electric power, the ice-making electric power, and the cold power can be expressed as follows:

$$
\begin{cases}
P_{Bcin}(t) = P_{Bc1in}(t) + P_{Bc2in}(t) \\
P_{Bc1in}(t) = 0 \\
P_{Bc2in}(t) = P_{Be2}(t) \cdot COP_{Be2}
\end{cases}
\tag{3}
$$

where: $P_{Bc2in}(t)$ is the input electric power (kw) of the electric chiller under the air-conditioning condition; $P_{Be2}(t)$ is the output cold power (kw) of the electric chiller; and $COP_{Be2}$ is the energy-efficiency coefficient of the ice-storage air-conditioning under the air-conditioning condition, which is the ratio of the input refrigeration capacity to the power consumption and characterizes the refrigeration performance of the ice-saving air-conditioning under the air-conditioning condition.

During the daytime, the cooling capacity required by the user is accomplished by both the chiller of the ice-storage air-conditioning and the melting of ice in the ice-storage tank. Within the adjustable interval, the refrigeration power interval allows for flexible allocation of ice-melting refrigeration capacity and chiller refrigeration capacity, taking into account the user's real-time cooling demand and the operating status of the ice-storage air-conditioning.

$$
\begin{cases}
Q_{Cooling}(t) = Q_{Cold}(t) + Q_{Ice}(t) \\
0 \le Q_{Cold}(t) \le Q_{Cold,\max}(t) \\
0 \le Q_{Ice}(t) \le Q_{Ice,\max}(t)
\end{cases}
\tag{4}
$$

where: $Q_{Cold}(t)$ and $Q_{Ice}(t)$ are the cooling capacity (kw·h) of the refrigerator and the ice-melting cooling capacity (kw·h) of the ice-storage air-conditioning at time $t$; $Q_{Cold,\max}(t)$ and $Q_{Ice,\max}(t)$ are the maximum cooling capacity (kw·h) of the refrigerator and the maximum cooling capacity (kw·h) of the ice-melting, respectively.

## 2.3. Energy Efficiency Ratio

The energy efficiency ratio is an indicator describing the efficiency of a refrigerator or ice machine, determined by the ratio of the output cooling capacity of the refrigeration unit to the input electricity, and the energy efficiency ratio can reflect the energy conversion efficiency of the system. According to the formula, the higher the ratio, the more cooling capacity will be converted per unit of electricity. Among them, the ice-making efficiency ratio of the ice-making machine of the ice-storage air-conditioning can be regarded as a constant because its working power is constant; however, the ice-making efficiency ratio $\eta_{EER}$ of the refrigeration machine is closely related to the load level, and its power will be adjusted in real-time according to the demand of the cold quantity so that the efficiency of the ice-storage air-conditioning has a significant difference between different loads. The energy efficiency ratio can be calculated by fitting the operating parameters of the refrigeration unit:

$$
\begin{cases}
\eta_{EET,t} = \eta_{EER} \times (c \cdot PLR_{PL,t}^2 + b \cdot PLR_{PL,t} + a) \\
R_{PL,t} = \dfrac{Q_{cold,t}}{Q_{cold,t,\max}}
\end{cases}
\tag{5}
$$

where: $\eta_{EER}$ is the rated energy efficiency ratio; *c*, *b*, *a* are the fitting coefficients; $R_{PL,t}$ is the load factor of the unit; $Q_{cold,t}$, and $Q_{cold,t,\max}$ are the hourly cooling capacity (kw·h) of the refrigeration unit and the upper limit of the cooling capacity (kw·h).

## 3. The Microgrid Modeling

### 3.1. Uncertainty in the Wind Power Output

Different wind speeds determine wind power outputs, where the uncertainty, volatility, and randomness of wind speeds affect the wind power output situation, thus leading to uncertainty in the wind power output. For this reason, wind power modeling needs to be carried out, considering the uncertainty model of its output.

The Wei-bull distribution function is used to simulate wind speed in engineering, and the specific expression is as follows:

$$f(v_{w,t}, r_{w,t}, c_{w,t}) = \frac{r_{w,t}}{c_{w,t}}\left(\frac{v_{w,t}}{c_{w,t}}\right)^{r_{w,t}-1} \exp\left[-\left(\frac{v_{w,t}}{c_{w,t}}\right)^{r_{w,t}}\right] \tag{6}$$

where: $v_{w,t}$ denotes the wind speed of the wind turbine impeller hub at time *t* at position *w*; $r_{w,t}$ and $c_{w,t}$ are the shape parameter and scale parameter of the wind turbine at time *t* at position *w*, respectively.

Considering that a constant power factor controls the wind power, the relationship between wind power output and wind speed at position *w* can be approximated by a segmented function:

$$P_w^{WTG} = \begin{cases} 0 & 0 \le V_w \le V_w^{ci} \\ P_w^{WTGn}\frac{V_w - V_w^{ci}}{V_w^r - V_w^{ci}} & V_w^{ci} \le V_w \le V_{wr}^r \\ P_w^{WTGn} & V_w^r \le V_w \le V_w^{co} \\ 0 & V_w^{co} \le V_w \end{cases} \tag{7}$$

where: $P_w^{WTG}$ is the rated capacity of the wind power at *w*; $V_w^{ci}$, $V_{wr}^r$ and $V_w^{co}$, are the cut-in, rated, and cut-out wind speeds of the wind power at wind farm *w*, respectively.

### 3.2. PV Output Uncertainty

Many factors affect the PV output profile, including solar light intensity, PV module surface temperature, and humidity. The variation of light intensity within a day is the main factor affecting the variation of PV output, and the beta distribution usually describes the variation of light intensity within a day.

$$f(I_{w,t}) = \frac{\Gamma(\alpha_{w,t} + \beta_{w,t})}{\Gamma(\alpha_{w,t}) + \Gamma(\beta_{w,t})}\left(\frac{I_{w,t}}{I_{w,\max}}\right)^{\alpha_{w,t}-1} \cdot \left(1 - \frac{I_{w,t}}{I_{w,\max}}\right)^{\beta_{w,t}-1} \tag{8}$$

where: $I_{w,t}$ and $I_{w,\max}$ are the light intensity and its maximum value at position *w*, respectively; $\alpha_{w,t}$ and $\beta_{w,t}$ are the two parameters of the beta distribution at position *w* at time *t*, respectively.

The PV output expression is as follows:

$$P_w^{PV} = \begin{cases} P_w^{pvn}\frac{I_w^2}{I_w^{stc}R_w^c} & I_w < R_w^c \\ P_w^{pvn}\frac{I_w}{I_w^{stc}} & I_w > R_w^c \end{cases} \tag{9}$$

where: $P_w^{pvn}$ denotes the rated output of the PV panel at *w*; $I_w$, $I_w^{stc}$ are the light intensity of the PV module at position *w* under actual conditions and test conditions, respectively; and $R_w^c$ denotes the point of light radiation at *w*.

### 3.3. Modeling Demand Response with User Comfort in Mind

Most previous studies have measured user comfort only from a temperature perspective and have ignored the impact of user comfort on users' actual participation in demand response decisions. Due to the incomplete dimension of the consideration factors, it is not easy to comprehensively reflect the user's comfort feeling in order to make the use of the ice-storage air-conditioning play out better the characteristics of peak shaving and valley filling. Taking into account that the user's perception of the indoor ambient temperature has a certain degree of ambiguity, this paper adopts the thermal sensation average scale predictive indicator (Predicted Mean Vote, PMV) to analyze the degree of sensitivity of the human body to the indoor temperature. The PMV indicator is a comfort equation that Prof. Fanger of Denmark proposed through many experiments [15]. It is based on the theory of thermoregulation and thermal comfort, which includes almost all human comfort factors.

$$
\begin{aligned}
I_{PMV} = &[0.303\exp(-0.036M) + 0.028] \\
&\{(M-W) - 3.05 \times 10^{-3}[5733 - 6.99(M-W) - p_a] - 0.42[(M-W) - 58.15] \\
&-1.7 \times 10^{-5}M(5867 - p_a) - 0.0014M(34 - t_a) - 3.96 \times 10^{-8}f_{c1}(t_{c1} + 273)^4 \\
&-(\overline{t_r} + 273)^4 - f_{c1}h_c(t_{c1} - t_a)\}
\end{aligned}
\tag{10}
$$

where: $I_{PMV}$ indicates the PMV indicator value; $M$ is the human metabolic rate, with static characteristics, taking the value of the range of [58, 100] W/m$^2$; $W$ is the mechanical work produced by the user; $p_a$ is the partial pressure of water vapor; $t_a$ is the air temperature, with dynamic characteristics, taking the value of the range of [22, 30] °C; $f_{c1}$ is the coefficient of clothing; $t_{c1}$ is the human body surface temperature; $\overline{t_r}$ is the average radiant temperature, with dynamic characteristics, taking the value of the range of [10, 40] °C; for the heat transfer coefficient [10, 40] °C; $t_c$ is the heat transfer coefficient.

PMV comfort index is a comprehensive index that is the influence of human comfort air temperature, humidity, flow rate, human clothing, activity status, and other multi-dimensional parameters under the joint effect of the integrated results of human comfort through the calculation of the human comfort level of the integrated quantification in the interval of [−3, 3]. By definition, the user feels most comfortable when the PMV is 0, and as the PMV value deviates more from 0, the human body feels less comfortable.

In order to reflect the relationship between user comfort indicators and temperature, and in order to satisfy the external factors in the range of comfort level, the PMV value depends on the temperature at different moments, and the relationship between the temperature PMV value ($\mu_{PMV}$) and the temperature T obtained at different moments is shown in Equation (11) [15]:

$$
\mu_{PMV} = \begin{cases} 0.3895(T - T_0) & T \geq T_0 \\ 0.4065(-T + T_0) & T < T_0 \end{cases}
\tag{11}
$$

When the indoor temperature is maintained at the time (in this paper, 26 °C), $\mu_{PMV} = 0$, the user's temperature comfort is the highest, indicating that the user's sensitivity to the temperature is low and has the most significant load regulation space during the cooling period. The recommended PMV, according to ref. [16], is in the range of [−0.5, 0.5], corresponding to indoor temperatures of 24.8 °C and 27.3 °C. The PMV of the room is in the range of [−0.5, 0.5] and [−0.5, 0.5].

## 4. Multi-Objective Optimal Scheduling Model for the Ice-Storage Air-Conditioning the Microgrid

### 4.1. Objective Function

This paper intends to characterize the ice-storage air-conditioning within the microgrid system to enhance the new energy consumption level and reduce the impact of wind power uncertainty on the power grid. The primary objective function is to minimize the operation and maintenance costs of each unit, maximize the generation rate of clean energy, and

minimize the cost of electricity for the air-conditioning users to improve the rate of new energy consumption and reduce the operation cost of the system.

Objective function I: minimize the operation and maintenance costs of each unit.

$$f_{objective} = \min \sum_{k=1}^{N_k} \beta_k \left( C_{rq,k} + C_{d,k} + C_{b,k} + C_{sloss,k} \right) + \varphi C_{VaR} \tag{12}$$

where: $\beta_k$ is the scenario probability; $\varphi$ is the risk coefficient; $C_{rq,k}$ is the cost of gas turbine gas purchase cost for scheduling in the microgrid under $k$ scenarios; $C_{d,k}$ is the cost of lithium battery storage system scheduling under $k$ scenarios; $C_{b,k}$ is the cost of the ice-storage air-conditioning during scheduling; $C_{sloss,k}$ is the cost of power lost in the distribution line under $k$ scenarios; and $C_{VaR}$ is the conditional value-at-risk of the scheduling result.

(1)　Operation and maintenance costs for miniature gas units.

$$C_{rq,k} = \sum_{t}^{24} \sum_{i}^{N_{MT}} \left( c_q f_{MT,t,k} \right) \tag{13}$$

where: $c_q$ is the natural gas price; $f_{MT,t,k}$ is the natural gas consumption per unit time of gas turbine at time $t$ in scenario $k$; $N_{MT}$ is the number of the gas turbines in the microgrid.

(2)　Operation and maintenance costs of the battery pack.

$$C_{d,k} = \sum_{t}^{24} c_d \left( P_{t,k}^d + P_{t,k}^c \right) \tag{14}$$

where: $c_d$ is the incentive price of the Li-ion battery storage system; $P_{t,k}^c$, $P_{t,k}^d$ are the charging power and discharging power of the Li-ion battery storage system at period $t$ in $k$ scenarios.

(3)　Operation and maintenance costs of the ice-storage air-conditioning.

$$C_{b,k} = \sum_{t}^{24} c_x P_{bd,t,k}^c x + c_z P_{bd,t,k}^s y \tag{15}$$

where: $P_{bd,t,k}^c$ and $P_{bd,t,k}^s$ are the electric power converted from the ice-making power and the cooling power of the ice-storage air-conditioning at time $t$ in the $k$-scenario, respectively; $c_x$ and $c_z$ are the incentive prices in different modes.

Ice-storage air-conditioning involves two types of incentive prices in the scheduling process: the first is in the consumption scenario, when there is an excess of distributed power output in the microgrid system, the ice-storage air-conditioning carries out the cold storage, and when there is a lack of power in the microgrid system, the electric power of the refrigeration machine is reduced. The cold storage device releases the cold to supplement the air-conditioning load, and at this time, the incentive price (denoted as the cost of the consumption of the ice-storage air-conditioning) is $c_x$, x = 1, y = 0. The second category is in the cutback scenario when the ice storage capacity of the storage unit is 0. High compensation is given because the power cutback affects the user's comfort, in which case the incentive price (denoted as the cutback cost of the ice-storage air-conditioning) is $c_x$, x = 0, and y = 1.

(4)　Distribution line loss power costs.

$$C_{sloss,k} = \sum_{t=1}^{24} c_m \left( \frac{P_{pcc,t,k}^2}{U^2} Z_1 \right) \tag{16}$$

where: $C_{sloss,k}$ is the power cost of distribution line loss under $k$ scenarios; $P_{pcc,t,k}$ is the contact line power at time $t$ under k scenarios; $U$ is the node voltage; $Z_1$ is the contact line impedance; $c_m$ is the power cost coefficient.

Objective Function II: Clean energy generation rate.

The clean energy generation rate refers to the percentage of clean energy generation, such as wind and solar, in the total system generation. In this paper's study, the clean energy consumed includes wind and photovoltaic generation.

$$\max F_2 = \frac{\sum\limits_{t=1}^{24} [P_{WT}(t) + P_{PV}(t)]\Delta t}{\sum\limits_{t=1}^{24} [P_{WT}(t) + P_{WT}(t) + P_{PV}(t)]\Delta t} \times 100\% \tag{17}$$

This optimization objective considers the environmental friendliness of the microgrid operation and is a substantial objective with a more significant objective value.

Objective Function III: Cost of electricity for the air-conditioning users.

$$
\begin{aligned}
f &= g - \max\{0, K_1\} \\
g &= \sum_{t=0}^{T} \left( \sum_{g=1}^{G} m_t P_z \right) \Delta t \\
K_1 &= \max\{0, l \cdot \Delta p\}
\end{aligned}
\tag{18}
$$

where: $g$ is the user's cost of electricity; $K_1$ is the power supply company's subsidy to the user's participation in regulation; the power supply company's subsidy to the user's cost is $l \cdot \Delta p$, $\Delta p$ is the air-conditioning user's participation in the regulation of the amount of regulation.

### 4.2. Constraints

Constraints I: Electrical Power Balance Constraints.

The system must follow the electrical power (active power) balance constraint. The entire system's power must meet the customer's electrical load demand at any given moment.

$$P_{MT}(t) + P_{WT}(t) + P_{PV}(t) + P_G(t) = P_{Ess}(t) + P_{Be1}(t) + P_{Be2}(t) + eload(t) \tag{19}$$

where: $P_{MT}(t)$ is $t$ time micro gas turbine power generation (kw); $P_{WT}(t)$ is $t$ time wind turbine output power (kw); $P_{PV}(t)$ is $t$ time photovoltaic unit output power (kw); $P_G(t)$ is the grid-connected operation of the microgrid and the grid exchange of power (kw), more significant than zero indicates that the power purchased from the power grid; less than zero indicates that the power is sold to the power grid; $P_{Ess}(t)$ denotes the power consumption of the battery at time $t$, whose value is greater than zero indicates that the storage of electricity, whose value is smaller than zero indicates the release of electrical power; $P_{Be1}(t)$ and $P_{Be2}(t)$, respectively, expressed in the ice-storage air-conditioning refrigerator refrigeration and storage of ice storage tank ice consumption of the electrical power at $t$ time.

Constraint II: Generation power constraint.

At all times, the generating power of micro gas, wind turbines, and photovoltaic units has maximum and minimum output limits.

$$
\begin{aligned}
P_{MT.\min} &\leq P_{MT}(t) \leq P_{MT.\max} \\
P_{WT.\min} &\leq P_{WT}(t) \leq P_{WT.\max} \\
P_{PV.\min} &\leq P_{PV}(t) \leq P_{PV.\max}
\end{aligned}
\tag{20}
$$

where: $P_{MT.\min}$ and $P_{MT.\max}$, respectively, is the minimum value and maximum value of the power output of the micro gas units in period $t$; $P_{WT.\min}$ and $P_{WT.\max}$, respectively, is the minimum value and maximum value of the power output of wind turbines in period $t$; $P_{PV.\min}$ and $P_{PV.\max}$, respectively, is the minimum value and maximum value of the power output of photovoltaic units in period t.

Constraint III: Wind and light abandonment constraints for wind turbines and photovoltaic units

$$0 < P_{WT}(t) \leq P_{WT.p} \tag{21}$$

$$0 < P_{PV}(t) \leq P_{PV.p} \tag{22}$$

where: $P_{WT.p}$ is the wind farm abandoned wind power, $P_{PV.p}$ is the photovoltaic power abandoned light power.

Constraint IV: The ice-storage air-conditioning constraint.

(1)  Capacity constraints.

At any given moment, the amount of cold stored in the ice storage tank shall not be less than the specified minimum amount of cold stored, nor shall it exceed the maximum amount of cold stored.

$$Q_{Cold.\min} \leq Q_{Cold} \leq Q_{Cold.\max} \tag{23}$$

(2)  Power constraints

At any given moment, the electric refrigeration power, the power of ice storage, and the ice release of the ice-storage air-conditioning have their power-limited range.

$$P_{Be.\min} \leq Q_{Be1}(t) \leq Q_{Be.\max} \tag{24}$$

$$P_{Bc2out.\min} \leq Q_{Bc2out}(t) \leq Q_{Bc2out.\max} \tag{25}$$

(3)  State constraints

Due to the continuous nature of the optimization cycle, the beginning and end cold storage capacity of the ice storage tanks should be kept almost constant from day to day.

$$|Q_{Cold}(24) - Q_{Cold}(0)| \leq \varepsilon_c \tag{26}$$

where: $\varepsilon_c$ is a small positive number. $Q_{Cold}(t)$ is the sum of $Q_{Cold}(t-1)$ stored in the ice storage tank at time $t-1$ and the real-time cold storage/release.

The amount of cold stored in the ice storage tank at time t is calculated by the following formula:

$$Q_{Cold}(t) = Q_{Cold}(t-1)(1 - \sigma_{Bc}) + [P_{Bc2in}(t)\eta_{Bc2in} - P_{Bc2out}(t)/\eta_{Bc2out}]\Delta(t) \tag{27}$$

where: $Q_{Cold}(t)$ and $Q_{Cold}(t-1)$ represent the cold capacity (kw·h) stored in the ice storage tank at time $t$ and time $t-1$, respectively; $\Delta t$ is the time interval, which is 1 h in the study of this paper; $P_{Bc2in}(t)$ represents the cooling power (kw) in the storage at time $t$; $P_{Bc2out}(t)$ represents the cooling power (kw) in the cooling release at time $t$; and $\sigma_{Bc}$, $\eta_{Bc2in}$, and $\eta_{Bc2out}$ represent the cold loss rate of the ice storage tank, the cooling storage efficiency, and the cooling release efficiency, respectively.

Constraint VII: Comfort Constraints.

User comfort is primarily temperature comfort. This paper uses PMV metrics to show the temperature range acceptable to the user when cooling so that the temperature value can be set rationally. During cooling, a comfortable range of indoor temperatures is set, and the relationship between the amount of cold released by the ice-storage air-conditioning and the indoor temperature is as follows:

$$\begin{cases} \frac{dT_t}{dt} = \frac{P_t^{reh}\Delta T + RT_t}{\rho} \\ T_{\min} \leq T_t \leq T_{\max} \\ \mu_{\min}^{PMV} \leq \mu^{PMV} \leq \mu_{\max}^{PMV} \end{cases} \tag{28}$$

where: $T_t$ is the value of indoor temperature at time $t$; $P_t^{reh}$ is the power released during the operation of the ice storage air conditioning; $\Delta T$ is the value of indoor temperature change; $R$ is the conductivity of the thermal resistance of the building materials; $\rho$ is the specific heat capacity of the air; $T_{\max}$, $T_{\min}$ are the maximum and minimum values of the indoor

temperature, the value of which is related to the selection of the degree of comfort; $\mu_{\min}^{PMV}$, $\mu_{\max}^{PMV}$ are the minimum and maximum values of the PMV indicators.

## 5. Multi-Objective Function Solving Based on the NSGA-III Algorithm

### 5.1. The NSGA-III Algorithm

At present, with the development of optimization technology, a large number of multi-objective intelligent optimization algorithms have been produced. Due to the different abilities of different algorithms to solve problems, some scholars have conducted comparative studies on some of the more widely used intelligent optimization algorithms. Song [17] compared the NSGA-II algorithm in the genetic algorithm with the multi-objective simulated annealing algorithm (MOSA) and the multi-objective particle swarm algorithm (MPSO) and pointed out that the NSGA-II algorithm not only converges faster but also converges to a high-quality solution that satisfies the optimization objective as well as the constraints. Ehghani [18] found that the NSGA-III algorithm outperforms the NSGA-II algorithm and the MPSO algorithm regarding convergence speed and runtime based on computational results from several numerical cases.

Both the NSGA-III algorithm and the NSGA-III algorithm can solve the multidimensional objective optimization problem [19]. However, due to the different selection mechanisms, the two algorithms perform differently regarding population diversity and convergence when solving three or more objective optimization problems. Deb [20] pointed out that the NSGA-II algorithm selects individuals by sorting the population through the crowding distance method, which performs better in solving a two-objective optimization problem. However, when solving three or more objective optimization problems, the solutions obtained are scattered in the non-dominated layer, which can easily cause the algorithm to fall into a local optimum. The convergence and diversity could be better. The NSGA-III algorithm, on the other hand, uses a set of predefined reference point locations, which ensures the diversity of the various clusters and also provides high convergence when dealing with multi-objective problems. For power networks, with the increase in the number of objectives and the increase in the size of the power network, it may not be easy to converge, and the solution accuracy is not high if the NSGA-II algorithm is still used. Therefore, more reliable and efficient optimization algorithms are needed to solve power networks' multi-objective decision optimization problems.

The NSGA-III algorithm is a more advanced algorithm that is further proposed based on the NSGA-II algorithm. The NSGA-III algorithm has the following advantages: (1) The NSGA-III algorithm uses fast, non-dominated sorting and an efficient selection strategy, which makes the algorithm search efficient and computationally efficient; (2) The NSGA-III algorithm enables the algorithm to improve the convergence and diversity of the population by introducing a reference point mechanism. It can guide the search direction so that the algorithm can adapt to different forms of objective functions and deal with multi-objective optimization problems with complex diversity. Based on this, the NSGA-III algorithm is used in this paper to solve the multi-objective optimal scheduling problem for the microgrid.

### 5.2. Steps for Solving the NSGA-III Algorithm

The NSGA-III algorithm solution flow is shown in Figure 2 with the following steps [19,20]:

(1)   Input an initial population P of size N and perform initialization operations on the relevant parameters, including the reference point, the number of iterations, the crossover and mutation probabilities, etc., and set the initial number of iterations to zero.

(2)   A crossover mutation operation produces a progeny population Q with a progeny population size of N.

(3)   Merge the parent population P and the offspring population Q to form a new population $R_t$, with a new population size of 2N.

(4) Based on the Pareto non-dominated ordering, the individuals in the new population $R_t$ are categorized into different non-dominated layers $R_t$, $F_2$, ..., $F_L$, to construct the new population $S_t$.

(5) Generate reference points and compute ideal points to construct a hyperplane through the set of ideal points in the new population $S_t$.

(6) Individual-associated reference points are used to select individuals based on small habitat retention operations until a population size that meets the conditions is filtered from $S_t$.

(7) Populations with a dominance rank of $R_{ank1}$ are selected at the end of each iteration to ensure that the Pareto maximum archive set is not exceeded.

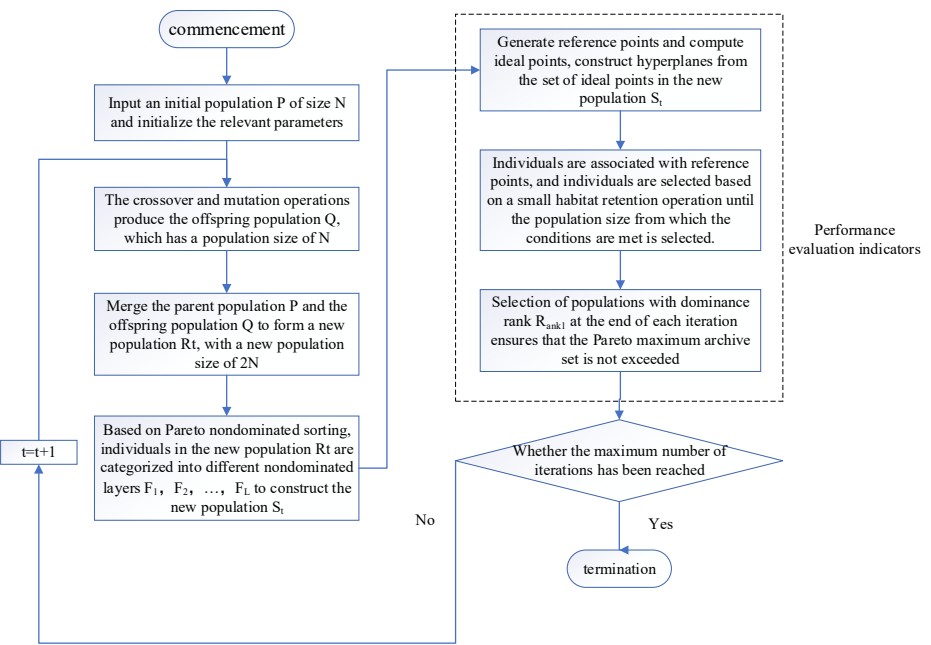

**Figure 2.** Steps of the NSGA-III algorithm solution.

In addition, the specific theories of reference point generation, standardized target space, correlation operations, and environment selection operations are detailed in the literature [19,21].

*5.3. Algorithm Performance Comparison*

This paper selects the DTLZ1 and DTLZ2 functions [22] as algorithm test functions to test the NSGA-II algorithm and the NSGA-III algorithm, respectively. The NSGA-II algorithm is compared to the proposed NSGA-III algorithm in terms of convergence and diversity metrics to arrive at an optimal solution algorithm adapted to the objective function established in this paper. The parameters are set as follows: The population size is N* = 100, the probability of variation is 1/D, and three groups of experiments are tested, respectively, with the dimensions of the objective function M* = 2, 3, and 5, and the results are shown in Table 1.

From Table 1, it can be seen that with DTLZ1 and DTLZ2 as the test functions, when the dimension of the objective function M* = 2, the NSGA-II algorithm and the NSGA-III algorithm have almost the same results in terms of convergence metrics and diversity metrics, and at this point, the two algorithms can be arbitrarily selected for solving; If the dimension of the objective function is 3, the NSGA-III algorithm has more minor results than the NSGA-II algorithm for both the convergence index and diversity index, which indicates that the NSGA-III algorithm has better performance than the NSGA-II algorithm when dealing with three-dimensional indexes; When the dimensionality of the objective

function is greater than 3, the results prove that the NSGA-III algorithm is suitable for dealing with high-dimensional multi-objective optimization problems.

**Table 1.** Comparison table of generational distances for evaluation metrics of multi-objective algorithms.

| M* | DTLZ1 | | DTLZ2 | |
|---|---|---|---|---|
| | **NSGA-III** | **NSGA-II** | **NSGA-III** | **NSGA-II** |
| 2 | 7.169 | 7.157 | 1.475 | 1.540 |
| 3 | 11.341 | 13.812 | 2.053 | 5.739 |
| 5 | 14.205 | 22.511 | 6.188 | 10.560 |

In order to better verify the performance and feasibility of the NSGA-III algorithm, the NSGA-III algorithm is compared and analyzed with the NSGA-II algorithm for Pareto's solution set accuracy. Parameter settings: initial population $N^* = 100$, objective function dimension $M^* = 3$. Figure 2 shows that, compared with the NSGA-II algorithm, the NSGA-III algorithm solves the microgrid optimal scheduling model with a higher Pareto and a more uniform distribution. The NSGA-III algorithm has a broader range of distribution, and most of the actual solutions are included. The NSGA-III algorithm not only effectively improves the rate of new energy consumption but also dramatically improves the economy of microgrid operation and users' power consumption (Figure 3).

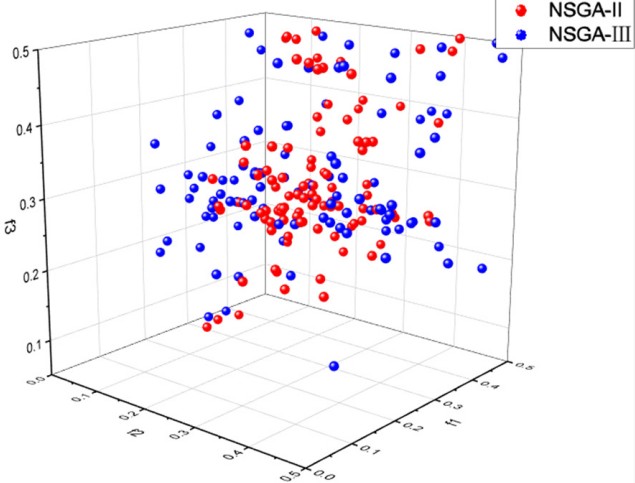

**Figure 3.** Comparison of Pareto optimal solution sets.

## 6. Example Analysis

### 6.1. Default Scenario Sets and Data

This paper uses Matlab 2023a to carry out simulation experiments to verify the feasibility and effectiveness of the ice-storage air-conditioning in participating in wind power consumption, as proposed in this paper, by setting up the microgrid system with or without the ice-storage air-conditioning. The example simulation parameters within the microgrid system are set as shown in Tables 2 and 3.

The wind and PV output curves and the cold load prediction curve within the microgrid system are shown in Figure 4.

From Figure 4, it can be seen that wind power and PV output fluctuate, and wind power is mainly concentrated in the period from 00:00 a.m. to 10:00 p.m., and wind power is at a low point in the afternoon; photovoltaic output is characterized by the opposite of wind power, with output at night in a trough period and peak output at midday; moreover, the user's cooling load demand is at its peak in the afternoon. The source-load imbalance within the microgrid system due to the stochastic nature of the output of distributed power

sources makes it necessary for the microgrid to frequently interact with the distribution grid to satisfy user demand, resulting in a lower overall operating economy.

**Table 2.** Parameters of equipment within the microgrid.

| Installations | Parameters (kw·h) |
| --- | --- |
| Wind power | 1500 |
| Photovoltaic power | 1000 |
| Gas turbine | 400 |
| Electricity storage power | 400 |
| The ice-storage air-conditioning storage cooling power | 800 |

**Table 3.** Operation and maintenance costs.

| Installations | Operation and Maintenance Costs ($/(kw·h)) |
| --- | --- |
| Wind turbine | 0.032 |
| Photovoltaic unit | 0.017 |
| Gas-fired unit | 0.033 |
| Accumulators | 0.007 |
| Electric coolers | 0.004 |
| Ice accumulator | 0.001 |

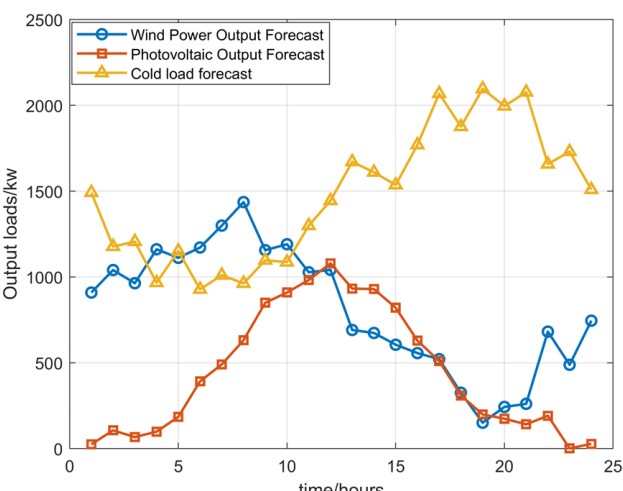

**Figure 4.** Forecast curves for wind, PV, and cooling loads.

### 6.2. The Microgrid Stochastic Optimal Dispatch Results

In this paper, the scenario method is adopted to describe the randomness and volatility of distributed power and cooling loads, and five typical scenarios are obtained, as shown in Figure 5.

In the case of the ice-storage air-conditioning participating in the microgrid scheduling, this paper makes the load demand in the microgrid match the distributed power output curve through the scheduling of equipment within the microgrid, such as the ice-storage air-conditioning, and the changes in the amount of abandoned wind and light after the ordinary air conditioners and the ice storage and the cold air conditioners participate in the stochastic optimal scheduling, respectively, are shown in Figure 6. As can be seen from the results of the scenario construction of wind turbines and cold loads in Figure 5, at 00: 00–10: 00, the wind power output reaches the peak, and in this period, the total output of distributed power is much larger than the load demand, and if no consumption measures are taken, all the excess power generation will be sent backward to the distribution network,

which will cause an impact on the grid to increase the loss of power, and the overall operating economy of the grid will be reduced. Decrease the ice-storage air-conditioning cold storage mode for energy storage to reduce the microgrid to the distribution network power backward transmission. Similar to the results of the PV unit and cold load scenario construction in Figure 5, the PV output peaks at 11: 00–15: 00, but since the user demand for the air-conditioning usage is higher at this time, there is no need to activate the ice-storage air-conditioning's cold storage or cold release modes. At 16:00–22:00, the load demand peaks; however, at this time, the output level of each distributed power source is low, and the air-conditioning loads need to be curtailed in order to satisfy the power balance within the microgrid system. At the same time, in order not to reduce customer satisfaction, the ice-storage air-conditioning stops storing cold in the chiller during this period. It uses the ice storage device to melt ice to supply cold (i.e., the cooling mode). From the above analysis, it can be found that the ice-storage air-conditioning's cold storage mode and cold release mode can be flexibly switched according to the changes in the output of the distributed power supply, which can be flexibly adjusted compared with ordinary air-conditioning.

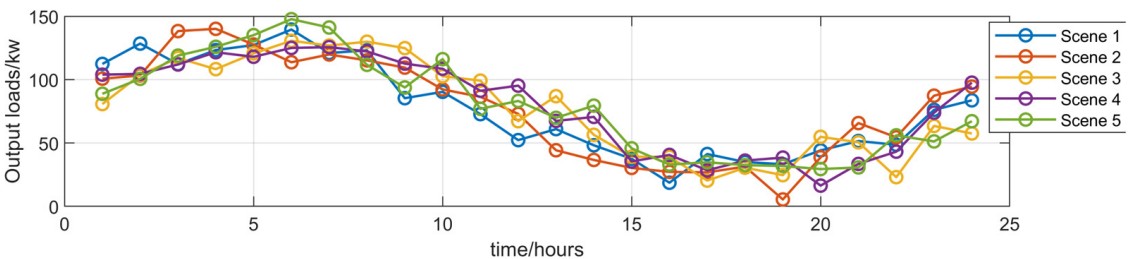

(**a**) Wind power scenarios

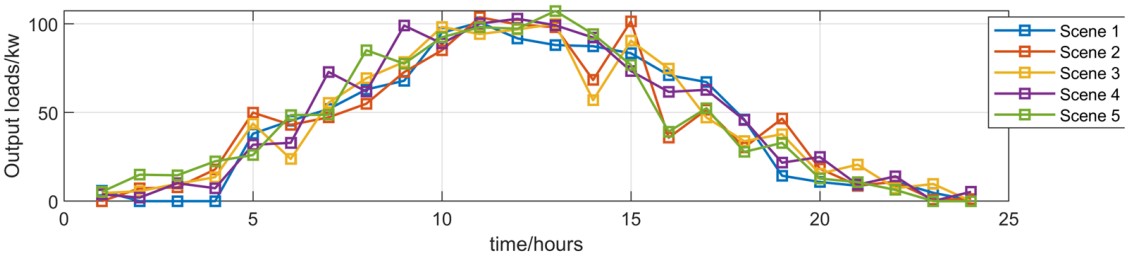

(**b**) Photovoltaic Output Scenario

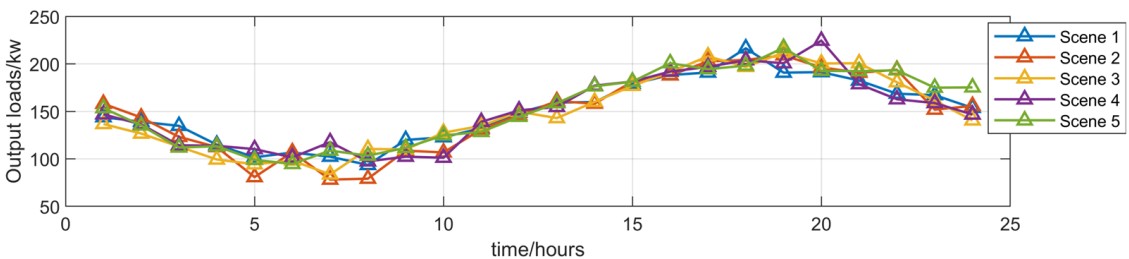

(**c**) Cold load scenarios

**Figure 5.** Results of constructing scenarios of wind turbine, PV, and cooling load outputs.

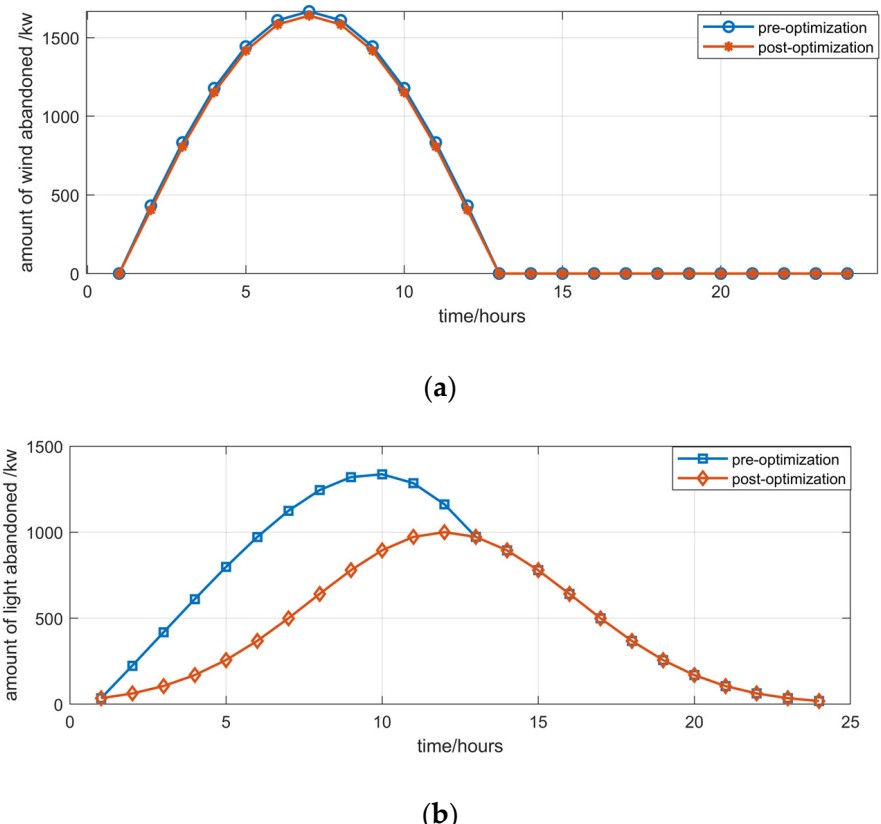

**Figure 6.** Comparison of wind turbine and photovoltaic unit changes in wind and light rejection when ice-storage air-conditioning is involved or not. (**a**) Wind turbine air abandonment during the ice-storage air-conditioning participation; (**b**) Photovoltaic (PV) units with the ice-storage air-conditioning participation in the amount of light discarded.

For the analysis of each piece of power equipment within the microgrid system, the comparison of the amount of wind and light discarded when the ice-storage air-conditioning participates in the microgrid optimal scheduling is shown in Figure 6. As can be seen from the figure, due to the participation of the ice-storage air-conditioning in the microgrid optimization scheduling, the amount of abandoned light from PV power generation has been substantially improved. However, the reduction of abandoned wind from wind turbine power generation is still relatively small. Refrigeration and air-conditioning are mainly applied in the summer, considering the characteristics of summer, which are generally full of light and less wind. Priority is given to the consumption of photovoltaic power generation. Hence, the ice-storage air-conditioning makes it possible to drastically improve the amount of discarded light from photovoltaic power generation. However, the impact on the amount of discarded wind from wind turbine power generation is relatively small.

*6.3. Economic Comparison of Ice-Storage Air-Conditioning Participation in Scheduling*

In this paper, a before-and-after comparison is made by setting the average value of the operating costs of the ice-storage air-conditioning with and without participation in the optimal scheduling of the microgrid, as shown in Figure 7. The operating characteristics of the ice-storage air-conditioning reduce the power reduction of the air-conditioning loads, which in turn reduces operational response costs. From Figure 7, the participation of the ice-storage air-conditioning in the microgrid optimal scheduling leads to a decrease in the operation and maintenance costs of the grid company, and the average value of the intraday cost is reduced by about 16.7%. At the same time, the participation of the ice-storage air-conditioning in the microgrid optimization scheduling also affects the user's electricity

cost, resulting in an average savings of 11.7% in the air-conditioning user's electricity cost, as shown in Figure 8.

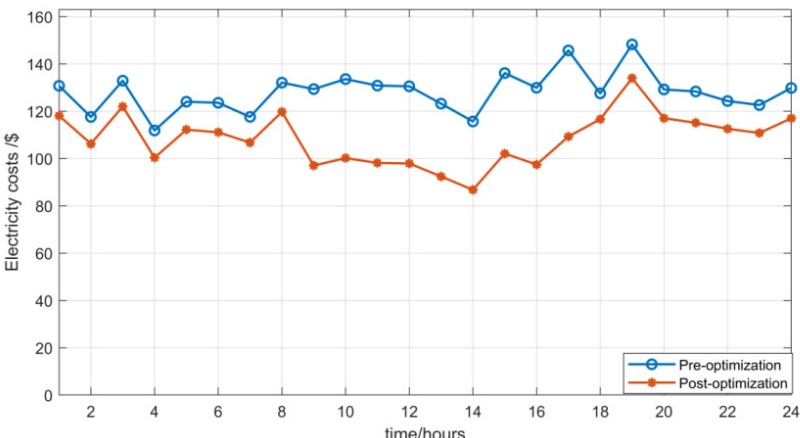

**Figure 7.** Comparison of operating costs before and after optimization of the ice-storage air-conditioning.

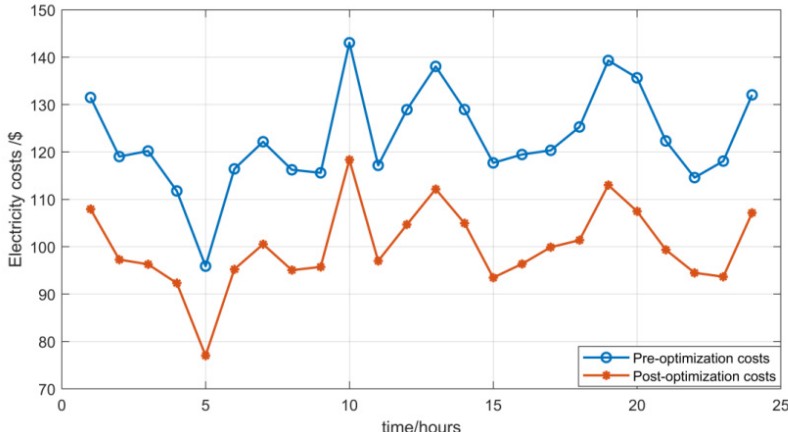

**Figure 8.** Changes in electricity costs for air-conditioning users before and after responding to regulation.

### 6.4. Taking User Comfort Effect Analysis into Account

When user comfort is not taken into account, the room temperature of the user's room is always maintained at 25 °C. When considering user comfort, due to the working characteristics of the ice-storage air-conditioning, by adjusting the power of the ice-storage air-conditioning at different times of the day so that the cold load can be flexibly matched with the output of wind power, the amount of new energy consumption can be effectively increased, and the operating cost of the equipment can be reduced. The change in indoor temperature before and after considering thermal comfort is shown in Figure 9.

In order to specifically analyze the impact of user comfort on the multi-objective optimization results, this paper sets four scenarios and compares four scheduling results, and the specific results are shown in Table 4.

Scenario 1: User comfort is not considered.

Scenario 2: Considering user comfort, μPMV = 0.5, r = 0.92.

Scenario 3: Considering user comfort, μPMV = 0.3, r = 0.92.

Scenario 4: Consider user comfort, μPMV = 0.5, r = 0.88.

From Table 4, it can be seen that the cooling load demand response considering user comfort has a positive impact on the wind energy consumption and operation cost of the ice-storage air-conditioning participating in new energy consumption. First, Scenario 1 improves wind energy consumption by 2.01 kW·h and reduces equipment operating costs by about $0.74 compared to Scenario 4. Second, when μPMV is constant, electricity comfort

is smaller, and when r is constant, temperature comfort is lower. At this time, the stronger the wind and light consumption capacity, the equipment operation economy is optimal.

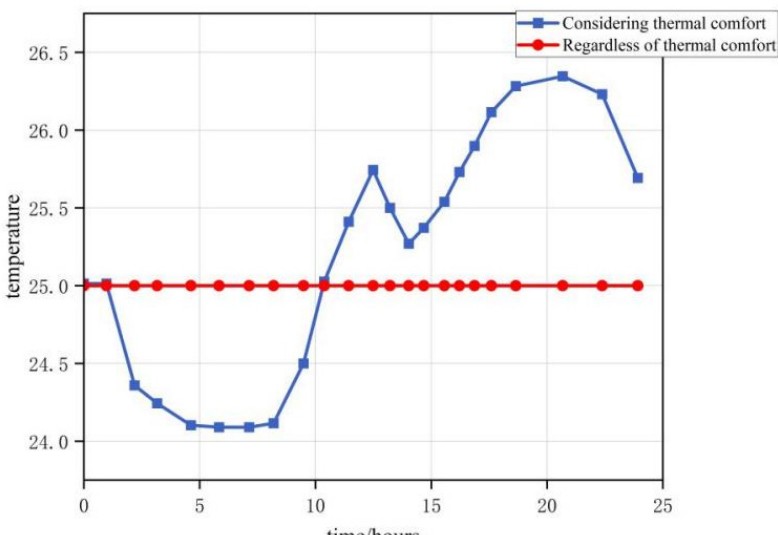

**Figure 9.** Change in indoor temperature before and after considering thermal comfort level.

**Table 4.** Analysis of the effect of different comfort levels on the participation of the ice-storage air-conditioning in wind energy consumption.

| Scenario | Wind and Solar Energy Consumption/(kW·h) | Equipment Operating Cost/$ |
|---|---|---|
| 1 | 326.73 | 132.31 |
| 2 | 328.71 | 131.72 |
| 3 | 328.59 | 132.01 |
| 4 | 328.74 | 131.57 |

## 7. Discussion

In the context of the new power system mainly based on new energy, for the problem of new energy consumption, this paper proposes a multi-objective optimization model and the NSGA-III algorithm solution algorithm considering the participation of the ice-storage air-conditioning in the wind and solar power consumption, comprehensively considering the optimization demand of the microgrid in the wind power and solar consumption rate, economy, and analyzes the operation mechanism of the ice-storage air-conditioning participating in the wind and solar power consumption, the NSGA-III algorithm, and the influence of the comfort level of users on the multi-objective optimization, and finally gets the following conclusions through the verification of examples. Analysis of the operation mechanism of the ice-storage air-conditioning participating in wind and solar energy consumption and the impact of the NSGA-III algorithm on multi-objective optimization.

(1) This paper establishes a multi-objective optimization model for the ice-storage air-conditioning to participate in wind and solar energy consumption, taking into account the issues of wind and light abandonment and the economy, and derives the optimal scheduling scheme with multiple objectives of the wind and solar energy consumption, operation cost, and user electricity cost to achieve the best comprehensive benefit, thus providing a reference for the work of scheduling and operation personnel. It can be seen through the analysis of the calculation example that the ice-storage air-conditioning participates in the optimal scheduling of the new energy microgrid, which has a significant effect on the promotion of the local consumption of photovoltaic but does not play a significant role in the consumption of wind power. Also, the

participation of the ice-storage air-conditioning had a significant impact on improving economic costs, with operating costs within the optimized microgrid reduced by approximately 10.5% and the cost of electricity used by the air-conditioning customers in response to regulation reduced by approximately 11.7%.

(2) Considering that demand response for user comfort improves system indicators, the next step can be to develop reasonable incentives to achieve a win-win situation between enterprises and users. Considering the influence of user comfort due to the working characteristics of the ice-storage air-conditioning, by adjusting the power of the ice-storage air-conditioning in different periods so that the cold load can flexibly match the output of photovoltaic, the amount of new energy consumption can be effectively improved, and the operating cost of the equipment can be reduced.

According to the simulation results of the example, it can be seen that, influenced by the summer season and user demand, the ice-storage air-conditioning participating in the optimal scheduling of the new energy microgrid mainly improves the consumption rate of photovoltaic power generation but has little influence on the consumption rate of wind power generation. In the future, consideration will be given to studying the impact of thermal storage electric heating and ice-storage air-conditioning on the rate of new energy consumption when they are jointly involved in the optimal scheduling of the new energy microgrid. In addition, the choice of parameters affects the algorithm's performance to a great extent, and the influence of the choice of parameters in the algorithm solution process needs to be considered in later studies.

**Author Contributions:** Conceptualization, Y.X., X.G. and J.H.; Writing—original draft, J.L. All authors have read and agreed to the published version of the manuscript.

**Funding:** This research was funded by Hubei Provincial Natural Science Foundation (2023AFB992) and Open Foundation of Hubei Key Laboratory for High-efficiency Utilization of Solar Energy and Operation Control of Energy Storage System (HBSEES202315).

**Institutional Review Board Statement:** Not applicable.

**Informed Consent Statement:** Not applicable.

**Data Availability Statement:** The data presented in this study are available on request from the corresponding author.

**Conflicts of Interest:** The authors declare no conflict of interest.

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
