# Peer review of "Optimized the Microgrid Scheduling with Ice-Storage Air-Conditioning for New Energy Consumption"

_sustainability, doi:10.3390/su16125133_

Round 1
Reviewer 1 Report
Comments and Suggestions for Authors
The paper is well-written overall, it can be accepted after implementing these improvements:
1. It is suggested to use unified format for Mathematical equations.
2. Performance of the algorithm highly depends upon the parameter selection. It is recommended to implement parameter selection methodologies such as DOE. Referred to article: Integrated planning and scheduling of multiple manufacturing projects under resource constraints using raccoon family optimization algorithm
3. Performance of algorithm is compared with the pre-optimization level. However, Performance of algorithm should also be compared with other existing algorithms based on multiple criteria.
4. While considering the multiple objectives, it is not possible to directly compare two methods. Author must use MCDM such as TOPSIS or other performance indicators such as IGD Referred to article: A smart algorithm for multi-criteria optimization of model sequencing problem in assembly lines; multi-policy deep reinforcement learning for multi-objective multiplicity flexible job shop scheduling
5. Conclusion of the paper is not well written. It should focus more on the result part instead of introduction.
6. Author must include the future directions.
7. The manuscript requires thorough proofreading for language and grammar. Conduct a detailed language check and proofreading before resubmitting.
Comments on the Quality of English LanguageThe manuscript requires thorough proofreading for language and grammar. Conduct a detailed language check and proofreading before resubmitting.
Author Response
Response letter
Dear reviewer:
Thank you for your decision and constructive comments on my manuscript. We have carefully considered the suggestion of the Reviewer and made some changes. We have tried our best to improve and made some changes to the manuscript. The yellow part has been revised according to your comments. Revision notes, paint-to-point, are given as follows:
- It is suggested to use unified format for Mathematical equations.
Response: We apologize for our careless mistake and thank you very much for the heads up. Based on your comments, we have changed the formula formatting issue to make it consistent throughout the manuscript.
- It is recommended to implement parameter selection methodologies such as DOE.
Response: Based on your suggestion, we have taken a serious look at the DOE methodology and in the process learned that DOE (Data On-site) is a design of experiments methodology used to explore and validate the effects of factors on outcomes. In DOE, experiments are usually divided into multiple combinations, each of which controls a factor and measures its effect on the outcome. In this way, it is possible to gain a more complete understanding of the effect of factors on the results and to identify the optimal combination of factors. The data in our manuscript are mainly analyzed and calculated based on the data derived from the modeling of distributed wind power output uncertainty and cold load uncertainty scenarios for micro-grids, and the influence of parameters on the performance of the algorithm is not considered, your comments are a new inspiration for us, but at this stage it may not be possible to realize it very soon, which will be one of the directions for our future research, and we will make up for it as soon as we have the condition to complete it, and publish the results.
- Performance of algorithm should also be compared with other existing algorithms based on multiple criteria.
Response: What we think is a terrific suggestion, we have changed the manuscript to add a comparative analysis of the NSGA II algorithm with the NSGA III algorithm already in use. The revised section is on page 13 of the manuscript in section 5.1.
- Author must use MCDM such as TOPSIS or other performance indicators such as IGD.
Response: What we think is a terrific suggestion, we have changed the manuscript to add a comparative analysis of the NSGA II algorithm with the NSGA III algorithm already in use. The revised section is on page 13 of the manuscript in section 5.1.
- Conclusion of the paper is not well written. It should focus more on the result part instead of introduction.
Response: Thank you very much for your comments, based on your suggestion, we have modified the conclusion section in the manuscript by adding the analysis of the results of the arithmetic simulation. The revised section is on page 18 of the manuscript in section 6.
- Author must include the future directions.
Response: Thank you very much for your comments, and as a result of your suggestions we have added a section to the manuscript on future directions. The revised section is on page 19 of the manuscript in section 6.
- The manuscript requires thorough proofreading for language and grammar.
Response: We apologize for the poor language of our manuscript. We worked on the manuscript for a long time and the repeated addition and removal of sentences and sections led to poor readability. We have scrutinized the contents of the manuscript and look forward to your criticism if there are still problems.
We have done our best to revise the manuscript and have highlighted the changes without affecting the content or framework of the manuscript. We express our sincere gratitude to the reviewers for their enthusiastic work and hope that the revisions will be recognized.
Once again, we sincerely thank you for your comments and suggestions.
Sincerely,
Corresponding author:
Name: Jiaxuan Li,
Email: lilililihuiqian@163.com
Reviewer 2 Report
Comments and Suggestions for Authors
The paper builds an optimal scheduling model for cold air condition and ice storage. Benefits of such scheme are well described and presented (via several scenarios) suggesting a new micro grid new energy consumption rate proving that ice storage air conditioning is more cost effective than traditional air conditioner.
Here are some comments and issues found into the manuscript I hope the authors will consider to address
1) Abstract page 1 / Summary page 17
While the abstract is well written, however I would probably underline one or two significant results with numbers. The same observation can be done about the last paragraph (it should be really named Summary and conclusions) where no effective number is described. When affirming “the overall ability to improve the operating economy of the microgrid system.” the logic question is “how much do we save? 10%, 20% ? Millions of dollars? Few cents? If overall efficiency is considered what is a “significant number” to be expressed? Please revise.
2) Section 2 / line 125
Please make sure all grammar errors / typos are eliminated as or example in “QCooling is the total cooling capacity (kwh)”. Note it should be kWh. Same problem is present later in Table 1 and Table 2.
3) Formula 1.5 / Line 166 – 167 – 168
In the text it is NOT clear ,as it seems that originally in the formula
that does NOT correspond to Rt but to RPL,t.
If they are NOT the same thing what are P and L?
4) Formula 216
Some formatting is needed here, as the formula is too much to the left. Please revise. I would also suggest for line 212 to write “…equation proposed by Fanger et al [16] though a large number….”
5) Discussion / Figure 4
Figure 4 is essentially NOT discussed at all. Why? There is a large difference between pre and post optimization in figure 4 b (PV), while the two trends seem almost the same for figure 4 a (Wind). Please do comment about it as it is NOT clear why this is correct or expected.
6) Discussion / Figure 6
The differences shown in this plot are very important. While it is obvious that timely differences cannot be the same, maybe it is worth to say what are: the minimum, maximum and average. These numbers would definitively give the reader a good idea of the percentage in costs savings. These results can be then recalled in the Conclusions.
In the present form the paper can be accepted with some minor reviews.

Few minor errors to be checked.
Author Response
Dear reviewer:
Thank you for your decision and constructive comments on my manuscript. We have carefully considered the suggestion of Reviewer and make some changes. We have tried our best to improve and made some changes in the manuscript. The yellow part that has been revised according to your comments. Revision notes, paint-to-point, are given as follows:
- While the abstract is well written, however I would probably underline one or two significant results with numbers.
Response: Thank you very much for your suggestion, it is very helpful to us, we have revised this part of the manuscript according to your comments, the revised section is in the abstract section on page 1 of the manuscript.
- Please make sure all grammar errors / typos are eliminated as or example in “QCooling is the total cooling capacity (kwh)”. Note it should be kWh. Same problem is present later in Table 1 and Table 2.
Response: We apologize for our careless mistake and thank you very much for the heads up. Based on your comments, we have made changes to both sections. We will try to improve them in the future to avoid making them again. Thank you for your understanding and support.
- In the text it is NOT clear ,as it seems that originally in the formula that does NOT correspond to Rt but to RPL,t. If they are NOT the same thing what are P and L?
Response: We apologize for our careless mistake and thank you very much for the heads up. Based on your comments, we have made changes to both sections. We will try to improve them in the future to avoid making them again. Thank you for your understanding and support.
- Some formatting is needed here, as the formula is too much to the left.
Response: We apologize profusely for our careless error and thank you very much for the heads-up. Based on your comments, we have made changes to the formula formatting issue to make it consistent throughout the manuscript. We will try to improve it in the future to avoid repeating it.
- Figure 4 is essentially NOT discussed at all.
Response: Your suggestion was very helpful to us, and we have revised this section of the manuscript based on your comments, the revised section is in the manuscript in section 5.3 on page 16.
- These numbers would definitively give the reader a good idea of the percentage in costs savings. These results can be then recalled in the Conclusions.
Response: Thank you very much for your suggestions, we have revised this section based on your comments, as well as the conclusion section, the revisions are in sections 5.4 on page 16 and 6 on page 18 of the manuscript.
We have done our best to revise the manuscript and have highlighted the changes without affecting the content or framework of the manuscript. We express our sincere gratitude to the reviewers for their enthusiastic work and hope that the revisions will be recognized.
Once again, we sincerely thank you for your comments and suggestions.
Sincerely,
Corresponding author:
Name: Jiaxuan Li,
Email: lilililihuiqian@163.com
Reviewer 3 Report
Comments and Suggestions for Authors
Dear authors,
thank you for your work.
Please, make IMRAD structure. Add Materials and Methods. Focus your Results and after give Discussion with research details and deep analysis.
Also you can make your Introduction more interesting.
for example,
Shushpanov, Ilia; Suslov, Konstantin; Ilyushin, Pavel; Sidorov, Denis N. Towards the Flexible Distribution Networks Design Using the Reliability Performance Metric // Energies , 2021, vol. 14, Issue 19, 6193
DOI10.3390/en14196193
Best regards, reviewer
Author Response
Dear reviewer:
Thank you for your decision and constructive comments on my manuscript. We have carefully considered the suggestion of Reviewer and make some changes. We have tried our best to improve and made some changes in the manuscript. The yellow part that has been revised according to your comments. Revision notes, paint-to-point, are given as follows:
Thank you very much for your suggestions, and we have revised the content according to your comments: Introduction corresponds to Section 1 on page 1 of the manuscript; Methods corresponds to Sections 2 and 3; Results corresponds to Section 5.1 on page 13 of the manuscript; Analysis corresponds to Sections 5.2-5.5 on page 13 of the manuscript; Discussion corresponds to Section 6 on page 18 of the manuscript; I don't know if my understanding is accurate, but if there is something wrong, I hope you can criticize it and look forward to your suggestions; Discussion section corresponds to section 6 on page 18 of the manuscript. I don't know whether my understanding is accurate or not, if there is something wrong, I hope you can criticize and correct me, and I very much look forward to your suggestions.
We have done our best to revise the manuscript and have highlighted the changes without affecting the content or framework of the manuscript. We express our sincere gratitude to the reviewers for their enthusiastic work and hope that the revisions will be recognized.
Once again, we sincerely thank you for your comments and suggestions.
Sincerely,
Corresponding author:
Name: Jiaxuan Li,
Email: lilililihuiqian@163.com
Round 2
Reviewer 1 Report
Comments and Suggestions for Authors
Author didn't answer the following questions
4- Author must use MCDM such as TOPSIS or other performance indicators such as IGD Referred to article: multi-policy deep reinforcement learning for multi-objective multiplicity flexible job shop scheduling
Comments on the Quality of English Languageit still needs proofread before publication. English can be further improved.
Author Response
Dear reviewer:
Thank you for your decision and constructive comments on my manuscript. We are very sorry that we did not fully understand your revisions in the first revision, and we have once again carefully considered your suggestions and made changes based on them. We have tried our best to improve and made some changes to the manuscript. The yellow parts have been revised according to your comments. The revision notes are as follows:
- Author must use MCDM such as TOPSIS or other performance indicators such as IGD.
Response: What we think is a terrific suggestion, we have added this section to the manuscript, and the changes are in section 5.3 on page 13 of the manuscript.
- The manuscript requires thorough proofreading for language and grammar.
Response: We apologize for the poor language of our manuscript. We worked on the manuscript for a long time and the repeated addition and removal of sentences and sections led to poor readability. We have scrutinized the contents of the manuscript and look forward to your criticism if there are still problems.
We have done our best to revise the manuscript and have highlighted the changes without affecting the content or framework of the manuscript. We express our sincere gratitude to the reviewers for their enthusiastic work and hope that the revisions will be recognized.
Once again, we sincerely thank you for your comments and suggestions.
Sincerely,
Corresponding author:
Name: Jiaxuan Li
Reviewer 3 Report
Comments and Suggestions for Authors
Dear authors,
you didn't read my previous recomendations.
Best regards, reviewer
Author Response
Dear reviewer:
Thank you for your decision and constructive comments on my manuscript. We are very sorry that we did not fully understand your revisions in the first revision, and we have once again carefully considered your suggestions and made changes based on them. We have tried our best to improve and made some changes to the manuscript. The yellow parts have been revised according to your comments. The revision notes are as follows:
Thank you very much for your suggestion, which is very helpful to us, and we have revised the content according to your comments. Regarding the Materials and Methods section, we have partially revised it, and our manuscript focuses on applying the NSGA III algorithm for multi-objective optimization solving, and the introduction of the NSGA III algorithm and the steps for solving it are in Section 5 on page 11 of the manuscript. For the results section, the revisions are in the manuscript in section 6 on page 14. Regarding the Discussion section, we have also analyzed it in more detail and added the content of our future research direction. The revisions are in the manuscript in section 7 on page 19. We do not know whether our understanding is accurate, and hope you can criticize and correct us if there is something wrong, and we are very much looking forward to your suggestions.
Once again, we sincerely thank you for your comments and suggestions.
Sincerely,
Corresponding author:
Name: Jiaxuan Li